# Non-traditional fluorescence in quadruple hydrogen bonded supramolecular polymers

Han Zuo[1], Yi Zeng[2], Qinghua Gao[3], Zexiang Wang[3], Qiannan Zhang[1], Youliang Zhu [4], Xiaoyan Zheng [2] ✉, Chuancheng Jia [3] ✉, Pingchuan Sun [1], Ben Zhong Tang [5] ✉ & Fenfen Wang [1] ✉

Non-traditional luminescent polymers exhibit significant advantages in bio-diagnostics and intelligent materials but suffer from low luminescence efficiency and limited functionality. Inspired by the excited-state proton transfer mechanism mediated by dense hydrogen bonds in jellyfish fluorescent proteins, we propose a strategy using dynamic quadruple hydrogen bonding ureidopyrimidinone motifs to create highly efficient luminescent polymers. By modulating the aggregated structure of the supramolecular units, proton transfer between paired motifs is activated, thereby achieving a high photoluminescent quantum yield up to 52% in supramolecular polyurethane. Ultrafast spectroscopy directly revealed this intermolecular proton transfer, while solid-state NMR spectroscopy confirmed the essential role of quadruple hydrogen bonds. The dynamically switchable hydrogen bonding structure endows the material with multifunctional integration, including strong fluorescence properties, high toughness, self-healing, reprocessability, and stimulus responsiveness. This research not only introduces a pioneering approach for advancing high-performance light-emitting materials but also enhances the prospects for their practical applications.

Non-traditional luminescent polymers (NTLPs) hold significant promise for bioimaging[1], anti-counterfeiting[2–4], and sensing[5–10], due to their advantages of good biocompatibility, film-forming ability, low cost, and facile synthesis. However, their practical adoption faces two fundamental constraints: (1) intrinsically strong non-radiative transitions arising from electronic localization and structural flexibility, leading to low photoluminescence quantum yields (PLQYs) compared to traditional conjugated systems; (2) the challenge of integrating high luminescence efficiency with multifunctionality for real-world applications[10–12]. Therefore, developing highly efficient NTLPs with integrated multifunctionality remains a pivotal challenge in light-emitting materials research.

Intriguingly, non-aromatic biomacromolecules widely present in nature, such as proteins and peptides devoid of any aromatic residues, have been shown to exhibit intrinsic fluorescence in the visible range[13]. This phenomenon provides critical insights into addressing the aforementioned challenges. Research demonstrates that the strong fluorescence of these bio-based NTLPs originates from the synergistic interplay between their precise high-density hydrogen bonding networks[14] and the excited-state proton transfer (ESPT) mechanism[15]. In particular, since the discovery of green fluorescent protein (GFP) in 1962, it has revolutionized the field of life sciences and medicine[16]. The amino acid residues around the chromophore in GFP create a rigid microenvironment with water molecules through hydrogen bonds,

[1]Key Laboratory of Functional Polymer Materials of Ministry of Education, Frontiers Science Center for New Organic Matter, College of Chemistry, Nankai University, Tianjin, China. [2]Key Laboratory of Photoelectroic/Electro-Photonic Conversion Materials, School of Chemistry and Chemical Engineering, Beijing Institute of Technology, Beijing, China. [3]Center of Single-Molecule Sciences, College of Electronic Information and Optical Engineering, Nankai University, Tianjin, China. [4]State Key Laboratory of Supramolecular Structure and Materials, College of Chemistry, Jilin University, Changchun, China. [5]School of Science and Engineering, Shenzhen Institute of Aggregate Science and Technology, The Chinese University of Hong Kong, Shenzhen (CUHK-Shenzhen), Guangdong, China. ✉e-mail: xyzheng@bit.edu.cn; ccjia@nankai.edu.cn; tangbenz@cuhk.edu.cn; wff@nankai.edu.cn

which accelerates the ESPT process and significantly enhances the fluorescence efficiency by suppressing the excited-state conformational relaxation[17,18] (Fig. 1a). ESPT is a photophysical process closely associated with fundamental life processes[19]. Luminescent materials based on ESPT have received extensive attention due to their good biocompatibility, large Stokes shifts, and stimulus responsiveness, and have been applied in fields such as fluorescence probes[20] and sensors[21]. However, replicating the kinetic characteristics and efficiency of intricate ESPT observed in biological systems during the molecular engineering of NTLPs has not been achieved yet.

In the conventional structural design of NTLPs, hydrogen bond interactions between groups like amide bonds[22,23] and urea bonds[24] can effectively enhance n-π conjugation and promote conformational rigidification, leading to improved fluorescence intensity. For example, recent reports have stated that enhanced fluorescence emission[25] and an extended spectral emission range[26,27] can be achieved by introducing high-density hydrogen bonds, which has become a promising strategy for improving fluorescence efficiency. Despite advancements, NTLPs face persistent bottlenecks: deficient synthetic access to efficient luminophores and inadequate mechanistic insight into structure-property relationships, which jointly hinder their development. Achieving a high photoluminescence quantum yield via precise hydrogen bond engineering in non-conjugated systems remains a critical challenge to be addressed.

Ureidopyrimidinone (UPy), a star molecule that dimerizes via strong quadruple hydrogen bonding, is widely used to construct supramolecular systems with high toughness and self-healing ability[28]. Recently, UPy has been employed as an auxiliary unit to enhance traditional fluorophores[29], or as an energy transfer medium to improve phosphorescence efficiency[30]. Additionally, photo-controlled hydrogen bonding UPy motifs enables reversible assembly processes[31]. However, its intrinsic luminescent properties and photoinduced keto–enol tautomerism have remained undiscovered until now.

Inspired by the multi-step proton transfer mediated by the hydrogen-bonding network confined within the β-barrel of GFP in Aequorea victoria, we propose a strategy of coupling ESPT with NTLPs containing high-density hydrogen bonding networks, as illustrated in Fig. 1b. This design enables successful preparation of an intrinsically highly luminescent polymer, utilizing a UPy-based supramolecular ESPT luminophore with programmable quadruple hydrogen bonds. The resulting supramolecular material (named PU·UPy) exhibits intense blue-violet photoluminescence with a remarkably high quantum yield of 52% among NTLPs reported to date. Through a combination of solid-state NMR, transient spectroscopy, and theoretical calculations, an ESPT mechanism is established as the origin of the strong emission. Incorporating UPy to polymer not only enhances mechanical properties, but also enabling multifunctional integration of efficient luminescence, self-healing, remoldability, and stimulus responsiveness. This polymeric material is well-suited for widely applications, including fluorescent coatings, cultural relic restoration, anti-counterfeiting, temperature/ion probes. This study not only establishes a paradigm for the molecular design of NTLPs but also develops a versatile material platform that integrates optical, mechanical, and intelligent response functionalities, paving the way for the next generation of bio-integrated devices and adaptive optoelectronics.

## Results and discussion
### Preparation and photophysical property of PU·UPy
The highly luminescent small molecule (UPy-DHDI) precursor and the supramolecular polymer (PU·UPy) incorporating UPy-DHDI were successfully prepared (Supplementary Figs. 1–6). The solid-state photoluminescence (PL) spectrum of PU·UPy (Fig. 2a) exhibits a consistent emission wavelength at 410 nm, independent of the excitation wavelength. This observation highlights the structural stability of its

chromophore and serves as a distinguishing feature when compared to conventional clusterization-triggered emission (CTE) phenomena observed in NTLPs. The ultraviolet (UV) absorption and excitation spectra of the material (Fig. 2b) indicate that the unconventional emission is associated with weak UV absorption in the range of 320 nm to 370 nm. This absorption can be attributed to the n-π* transition of the UPy heterocycle, which was not observed in either the solution absorption spectrum of PU-UPy or the solid-state UV absorption spectrum of the blank PU sample (PU-b) (Supplementary Fig. 7). These findings suggest that the absorption in the range of 320–370 nm originates from aggregation-induced effects of UPy motifs. More importantly, PU·UPy exhibits a remarkably high PLQY of 52% in the solid state (Fig. 2c), surpassing most reported NTLPs (Fig. 2d)[8,22,23,25,32–36]. Due to the presence of multiple hydrogen bond interactions, PU·UPy exhibits high tensile stress (~20 MPa) and high stretchability (965%) (Supplementary Fig. 8). Compared to the blank PU sample, PU·UPy demonstrates significantly enhanced mechanical strength, toughness, and PLQY (Fig. 2e), thereby providing a basis for achieving multi-functional integration in polymeric materials. Circularly polarized fluorescence spectroscopy indicates a clear, albeit weak, left-handed polarization in the fluorescence emission of PU·UPy (Fig. 2f).

The luminescent properties of the PU·UPy were further investigated in DMF and CHCl₃ with various concentrations (Fig. 2g–i), and the absorption spectra (Supplementary Fig. 7) were measured. It was found that the trend of the emission intensity changing with concentration was inconsistent in the two solutions. Notably, in a $10^{-3}$ M CHCl₃ solution (Fig. 2g), the PL spectra of the sample transition from single-peak emission to double-peak emission. Solvent dependence and multi-peak emission are significant characteristics of ESPT[26,32]. Due to the existence of a high-density dynamic proton network between UPy dimers, proton transfer is facilitated efficiently. Therefore, it is speculated that the unique fluorescence properties of PU·UPy are related to the ESPT behavior facilitated by quadruple hydrogen bonds.

### The luminescence mechanism of UPy motifs
To further uncover the underlying luminescence origin of the UPy motifs, we first investigated the fluorescence properties of the monomer UPy-DHDI used to construct the supramolecular polymer. It was found that UPy-DHDI not only exhibited emission characteristics and fluorescence quantum yield values highly consistent with those of the polymer material, but also demonstrated consistency in its excitation spectrum compared to the polymer (Supplementary Fig. 9a, c). Combined with the analysis of the solid-state UV absorption spectrum, it was found that the changes in the excitation behavior of both originated from the enhanced absorption in the range of 320–370 nm (Supplementary Fig. 10). This result directly confirms that the UPy motif is the fundamental structural unit responsible for the luminescent functionality of the PU·UPy. It has been reported in the literature that the existence of the inherent tautomerization of heterocyclic ring-based system, such as UPy derivatives[37]. And the structure control of intramolecular proton tautomerism has been an inspired and fruitful research area in supramolecular chemistry[38]. In fact, the equilibrium of proton tautomer of UPy is solvent and temperature dependent[39], referring to a permanent conformational change in the ground state. Based on the experimental results presented above as well as the existence of the inherent tautomerization of UPy, we hypothesize that UPy undergo photoinduced tautomerism in the excited state. Thus, a luminescence mechanism for UPy based on this inference was proposed: in the ground state, the UPy dimer usually has a donor-donor-acceptor-acceptor (DDAA) hydrogen-bonding array with keto form; after transitioning to the excited state, the UPy dimer undergoes an intermolecular double proton transfer to form an enol form (Enol₁) with a DADA hydrogen-bonding array, and may further isomerize to another enol form (Enol₂) with an ADDA hydrogen-bonding array (Fig. 3a).

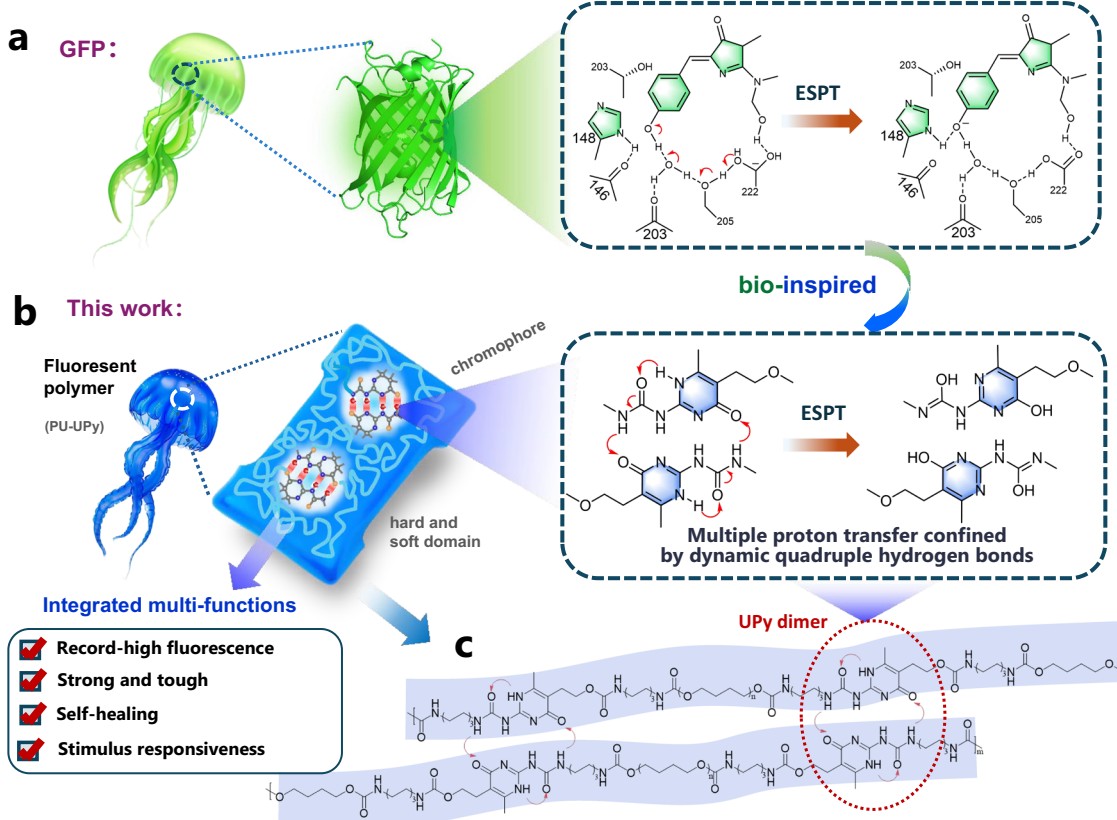

**Fig. 1 | Bio-inspired design of the multi-functional fluorescent supramolecular polyurethane. a** Diagram of the ESPT mechanism in GFP. **b** Diagram of the fluorescence mechanism of PU-UPy. **c** Polymer chain structure of PU-UPy (the fluorescent unit is marked with a red dotted line).

Systematic theoretical investigation using time-dependent density functional theory (TD-DFT) was performed to verify the aforementioned mechanism. For simplification of the calculations, UPy-DHDI was employed as a model molecule instead of PU-UPy. Structural optimization was conducted at the B3LYP/6-31G* level of theory, utilizing the Gaussian 16 and Multiwfn software packages[40,41]. The calculation results show that proton-transfer-driven tautomeric transformation in the excited state of UPy dimer could cause a red-shifted emission. The fluorescence emission wavelengths can be predicted by calculating the vertical excitation energies from $S_0$ to $S_1$ state of three tautomeric forms of UPy-DHDI, which are 3.34, 3.01, and 2.87 eV, respectively (Fig. 3b). It was found that the enol form has lower energy than the keto form, and the energies of the three tautomeric forms decrease progressively. This confirms that the proton transfer and tautomerization processes are thermodynamically favorable. The predicted maximum absorption wavelengths are 371.8, 411.5, and 431.3 nm for Keto, $Enol_1$ and $Enol_2$ form of UPy, respectively. Notably, the calculated wavelength of $Enol_1$ exhibits excellent agreement with the emission wavelength obtained from the fluorescence spectrum (Fig. 2a). This suggests that the formation of $Enol_1$ structure in the excited state is responsible for the efficient luminescence of UPy motifs. Moreover, the emission wavelength of $Enol_2$ form is calculated as 431.3 nm, which corresponds to the long-wave emission peak of PU-UPy in CHCl₃ (peak II in Fig. 2h), providing evidence for the existence of $Enol_2$ form in the excited state. Potential energy curves (PECs) can effectively reveal the feasibility and energy barriers of proton transfer in both ground and excited states. The ground- and excited-state PECs for the proton-transfer product in the first-step were calculated (Supplementary Fig. 11a, b). The results show that the energy barrier for proton transfer in the excited state is significantly reduced compared to that in the ground state, thereby further demonstrating the

feasibility of ESPT rather than intermolecular proton transfer in the ground state.

To obtain direct experimental evidence for proton transfer dynamics, femtosecond transient absorption (fs-TA) spectroscopy was employed to investigate the ultrafast photophysical behavior of PU-UPy in the excited state. The transient absorption optical path was pre-assembled to meet the testing needs (Supplementary Figs. 12 and 13). Following excitation at 380 nm (derived from a 1030 nm Pharos femtosecond laser), two distinct spectral features emerged within the first 20 ps of the transient absorption spectra (Fig. 3c). A short-wavelength signal attributed to the pre-transfer keto form decays rapidly on the sub-picosecond timescale. Simultaneously, a long-wavelength signal assigned to the post-transfer $Enol_1$ form rises and maximizes at -1 ps. These temporal-spectral characteristics provide compelling evidence for ESPT[42], unambiguously confirming that UPy units undergo multi-proton transfer in the excited state, accompanied by a keto → enol tautomerization. The decay kinetics of both signals reflect the relaxation of excited molecules to the ground state (Fig. 3d). Time-resolved fluorescence spectroscopy based on time-correlated single photon counting (Supplementary Fig. 14), was employed to capture the weak emission signal from $Enol_2$ with a long delay time. The evolution of the emission spectra with delay time demonstrates the existence of the ESPT process and the existence of the $Enol_2$ form. The energy transfer process of the excited state of PU-UPy is summarized with two ESPT processes (Fig. 3e). The foregoing experimentally resolved ESPT dynamics perfectly corroborates the theoretically predicted multi-proton transfer pathway confined by multiple hydrogen bonds, thereby establishing a comprehensive microscopic mechanism: "ESPT → tautomerization → fluorescence enhancement". To more accurately describe the proton transfer process, we performed potential energy surface scans for both the intramolecular and

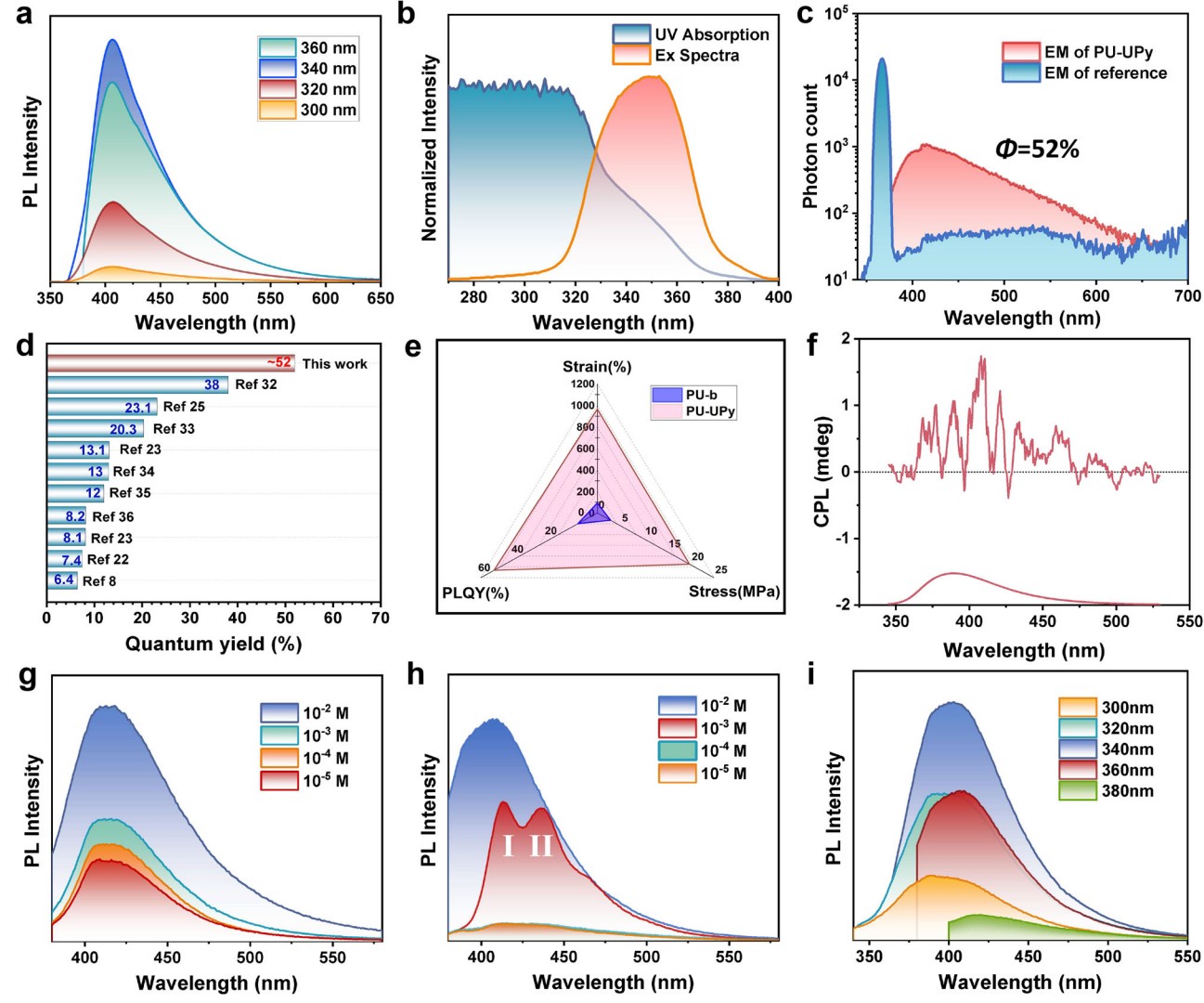

**Fig. 2 | Nonconventional fluorescence properties of PU-UPy.**
**a** Photoluminescence spectra of PU-UPy film sample (excited at different wavelength). **b** Normalized UV absorption and excitation spectra of PU-UPy (Solid state). **c** PLQY of PU-UPy film measured using an integrating sphere. The PLQY value is 52% upon excitation at 365 nm. **d** QY comparison of PU-UPy with some non-traditional fluorescence systems reported. **e** Radar chart of PU-UPy compared to blank sample PU-b. **f** CPL spectra of PU-UPy film. **g, h** PL spectra of PU-UPy in DMF and CHCl$_3$ with different concentration. I and II represent the emission peaks corresponding to different structures. **i** Excitation-dependent PL spectra of PU-UPy in DMF (10$^{-3}$ M).

intermolecular proton transfer pathways (Supplementary Fig. 15). The results suggested that direct intermolecular proton transfer from the keto configuration faces a high energy barrier (2.97 eV), making it unlikely under conventional photoexcitation. In contrast, the intramolecular pathway has a much lower barrier (1.12 eV), suggesting it occurs preferentially. Once the intramolecular transfer is complete, the resulting electronic rearrangement significantly reduces the barrier for subsequent intermolecular transfer, facilitating a concerted mechanism. Therefore, the ESPT of UPy dimers should be a intramolecular-intermolecular cooperative multi-proton transfer process.

### The influence of UPy motif structure on luminescent properties

The photophysical properties of the luminescent unit are closely tied to its aggregated structure[43,44]. To investigate the structure-function relationship of emissive UPy motifs, a control sample named UPy-DHI was synthesized by replacing the HDI reagent with hexyl isocyanate. This modification yielded a weakly emissive sample (Fig. 4a). Spectral characterization revealed that UPy-DHI exhibited a quantum yield below 10% and displayed excitation wavelength-dependent fluorescence emission (Supplementary Fig. 9b, d). UPy-DHI exhibited fixed

emission wavelength and distinct emission peaks in CHCl$_3$ (Supplementary Fig. 18a, b), similar to those of UPy-DHDI in the solid state. These findings indicate that changing the aggregated structure of UPy-DHI can endow it with a luminescent behavior similar to that of UPy-DHDI. We further explored the differences in the aggregate structures of the two samples. X-ray diffraction results reveal that UPy-DHI exhibits a crystalline structure, whereas UPy-DHDI shows disordered amorphous structure (Fig. 4b). Subjecting the UPy-DHI sample to liquid nitrogen quenching produced an amorphous material (Supplementary Fig. 16) that still exhibited weak fluorescence (Supplementary Fig. 17), confirming that crystallization is not the origin of the fluorescence differences. Furthermore, one-dimensional profiles of small-angle X-ray scattering experiment show a distinct peak at $q \sim 1.33$ Å$^{-1}$, corresponding to a d-spacing of 4.72 Å, close to typical π-π stacking distances, indicating plane separation formed by UPy dimers. In contrast, no significant scattering peaks were observed in the wide-angle region for UPy-DHDI; only a very weak shoulder peak was present at $q \sim 1.24$ Å$^{-1}$ (d-spacing ~5.06 Å). This indicates an increased interplanar distance and a disruption of ordered packing (Fig. 4c). DFT simulations revealed the tetrameric structures of UPy-DHI and UPy-DHDI

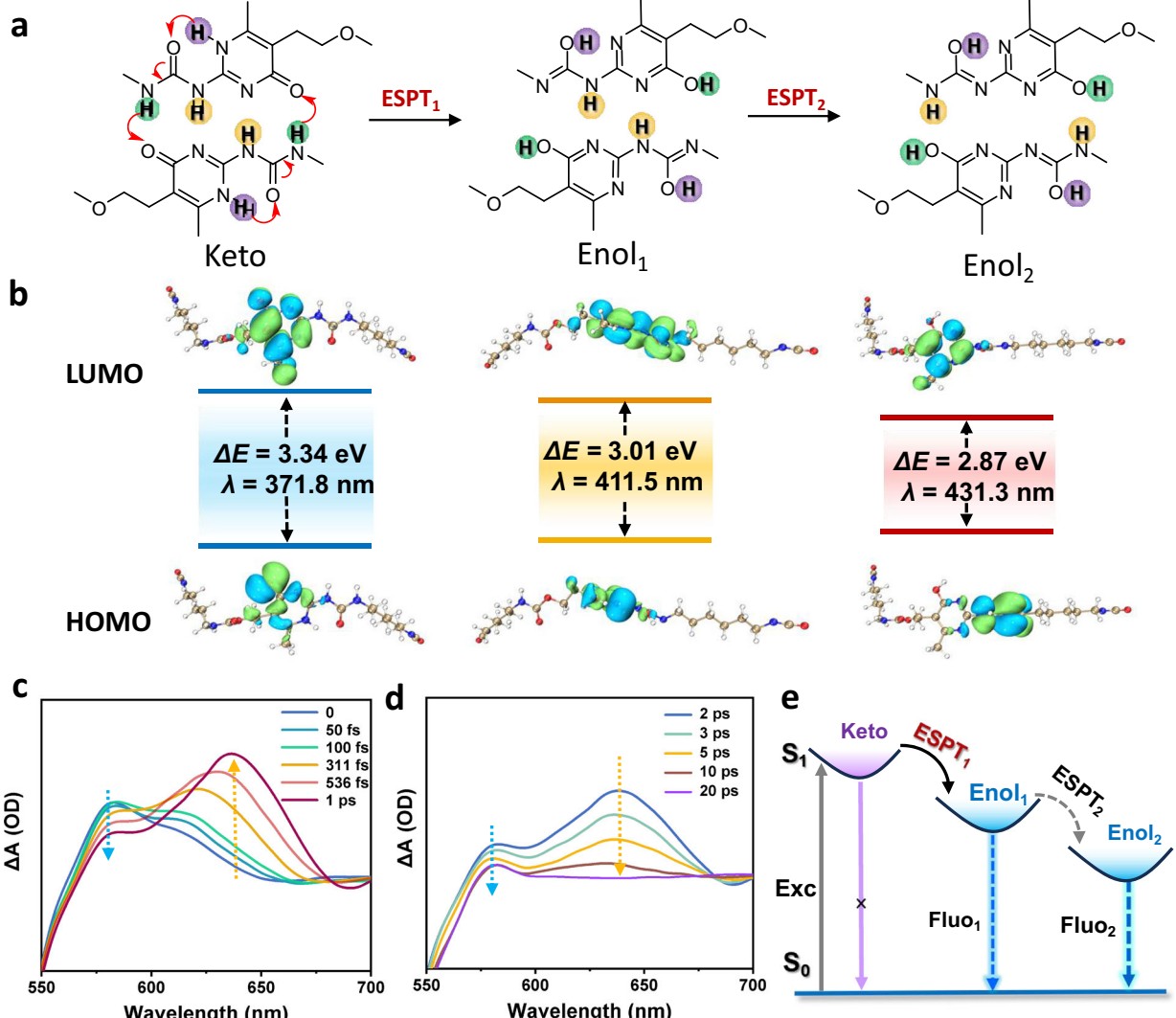

**Fig. 3 | Excited state ultrafast dynamics of UPy dimers. a** ESPT process of UPy dimers. **b** The energy gap of three tautomeric forms of UPy-DHDI at S1 state by TD-DFT simulation. **c**, **d** Time-resolved transient absorption of PU-UPy film monitored by pump–probe setups, a Pharos laser with a center wavelength of 1030 nm was used. **e** The photophysical processes of UPy groups.

(Supplementary Fig. 19a, b). The results indicate that UPy-DHDI tetramers exhibit a longer interplanar distance than those of UPy-DHI, consistent with the WAXS results. The increased molecular spacing and more disordered molecular orientation directly weaken intermolecular π–π stacking. This structural change enhances fluorescence efficiency through two distinct mechanisms. First, by reducing the number of π–π stacking structures that can act as exciton quenching sites, it ensures the efficient progression of the ESPT process. Second, the suppression of π–π stacking mitigates aggregation-caused quenching, thereby preventing the dissipation of ESPT-generated fluorescence via non-radiative pathways.

$^1$H double-quantum/single-quantum (DQ/SQ) chemical shift correlation NMR spectrum is a powerful tool for elucidating the packing structure and hydrogen bonding interactions of molecules[45]. From the proton 2D DQ/SQ spectra (Fig. 4d, e), the presence of DQ correlation signal $H_b$-$H_c$ indicates that both UPy-DHDI and UPy-DHI formed a DDAA-type hydrogen-bonding array, further supporting the correctness of above hypothesis regarding the mechanism of emission (Fig. 3a). Notably, the $^1$H NMR spectrum of UPy-DHI—both crystalline and amorphous—shows a distinct signal for the isolated proton $H_c'$, which is not involved in UPy hydrogen bonding (Fig. 4e and

Supplementary Fig. 20). This directly demonstrates the disruption of the quadruple hydrogen-bonding network in UPy-DHI, which we proposed suppresses the intermolecular proton transfer process, leading to its markedly weaker fluorescence. When dissolved in CHCl₃, the emission behavior of UPy-DHI undergoes a significant change and closely resembles that of UPy-DHDI and PU-UPy in the solid state. The above experiments confirmed that the aggregated state structure can modulate the generation of UPy fluorescence by influencing hydrogen bond formation and stacking structure. Furthermore, the $^{13}$C NMR relaxation spectra (Supplementary Fig. 21 and Table 1) shows that the spin-lattice relaxation times ($T_1$) of individual carbon atoms within the UPy-DHDI structure are shorter than those in UPy-DHI, which implies that the chain in UPy-DHDI are more mobile than those in UPy-DHI. This observation suggests that the commonly held belief that restricted molecular motion leading to enhanced luminescence is not the primary mechanism underlying UPy luminescence. Furthermore, these results provide an explanation for the highly efficient fluorescence of UPy-DHDI from the perspective of conformational freedom. The relatively amorphous structure offers greater conformational flexibility, thereby allowing dynamic rearrangement of the hydrogen-bonding network, which is crucial for the ESPT process.

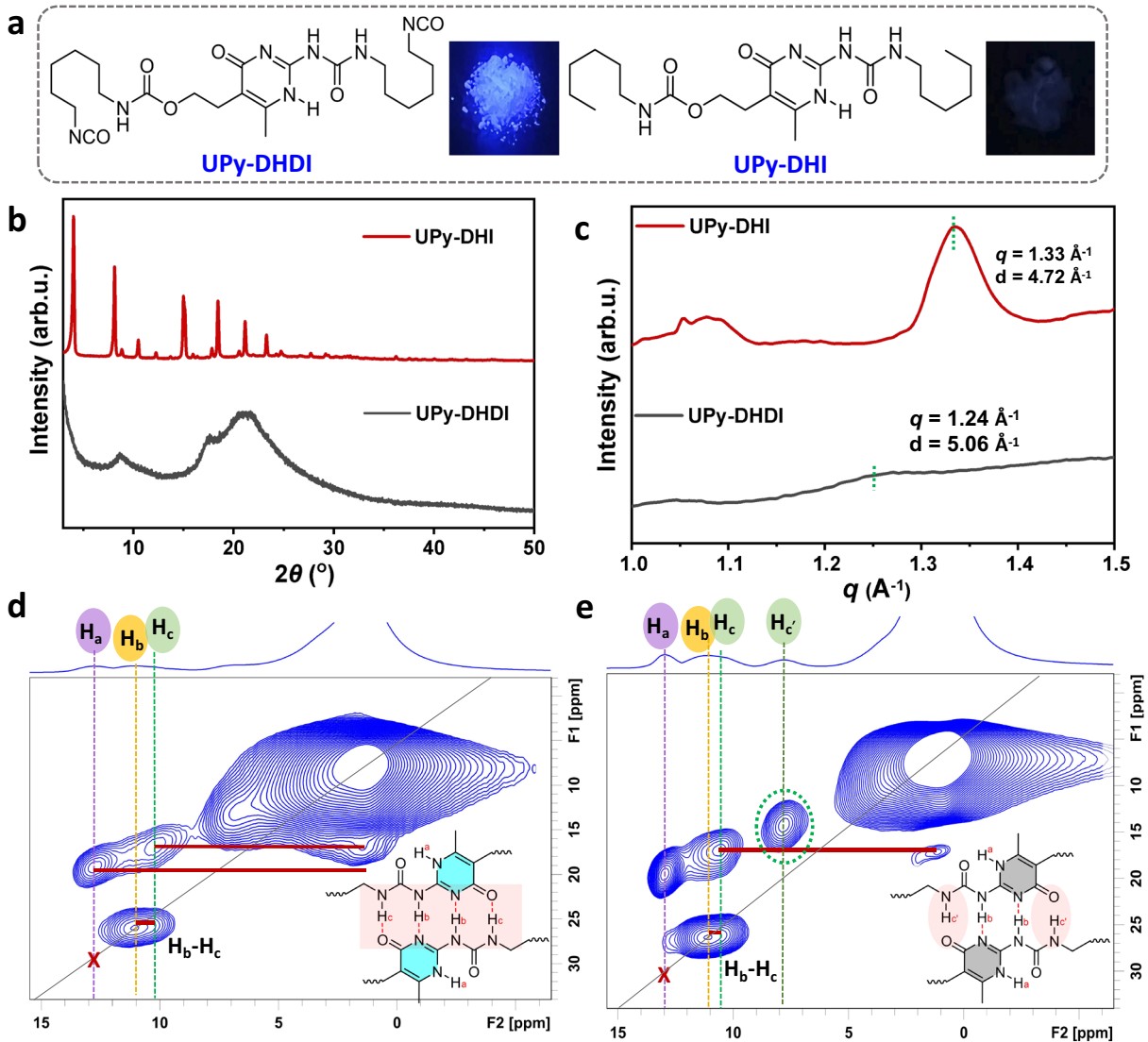

**Fig. 4 | Aggregated structure of UPy with various end groups. a** Molecular structure of UPy-DHDI and UPy-DHI (inset photograph of samples irradiated with a 365 nm UV lamp). **b** XRD spectra of the above samples. **c** WAXS spectra of the above samples. **d**, **e** Solid-state NMR two-dimensional DQ/SQ correlation spectra of the above two samples. The pulse sequence for this experiment is shown in Supplementary Fig. 15.

## Molecular dynamics (MD) simulations for aggregated structure of UPy dimer

MD simulations were used to study the aggregated structure of UPy-DHDI dimers and tetramers, revealing the influence of aggregated structure of chromophores on fluorescent properties. The results show that both the dimer and tetramer form stable quadruple hydrogen-bonding arrays (Fig. 5a). The tetramer exhibits irregular stacking between planes, which suppresses exciton annihilation and resonance energy transfer in the excited state. Hydrogen bond distributions during assembly were quantified: approximately 4 bonds in the dimer and 8−9 in the tetramer (Fig. 5c and f), confirming consistent formation of stable quadruple hydrogen-bonding arrays. Calculations of interplanar distances and dihedral angles within the tetramer (Fig. 5d−g) show that one pair of UPy motifs (UPy1-UPy2) has a dihedral angle <30° and an interplanar distance of 8 Å, while the other pair (UPy3-UPy4) has a smaller interplanar distance but a dihedral angle >30°. These results indicate irregular stacking between UPy dimer units with quadruple hydrogen bonding, which can ensure efficient ESPT and enhances luminescence emission[46]. This conclusion is also supported by SAXS and DFT simulation results previously presented. On the basis of the above discussion, a comprehensive

understanding of the luminescence mechanism similar to that observed in jellyfish fluorescent protein (Fig. 1) of PU-UPy polymer can be achieved. Within this mechanism, the UPy dimer undergoes concerted intra- and intermolecular multiple proton transfer upon excitation. The resultant tautomer enables red-shifted visible-light emission. Crucially, the rigid end groups (or polymer chains engaged in hydrogen bonding) prevent the formation of regular interplanar stacking structures. This suppression occurs without hindering proton transfer, thereby preventing the efficiency losses caused by exciton annihilation and resonance energy transfer.

## Application of strong emissive PU-UPy

Conventional fluorescent materials often suffer from functional limitations, with integration of luminescence and other essential properties proving challenging. In this work, this challenge is addressed by activating the intrinsic fluorescence of UPy motifs, achieving simultaneous integration of high toughness, self-healing, and multi-stimuli responsiveness in a UPy-based supramolecular polymer. This synergistic multifunctionality adapts to diverse applications, as demonstrated by comprehensive performance assessments that highlight cross-disciplinary potential. PU-UPy exhibits excellent optical

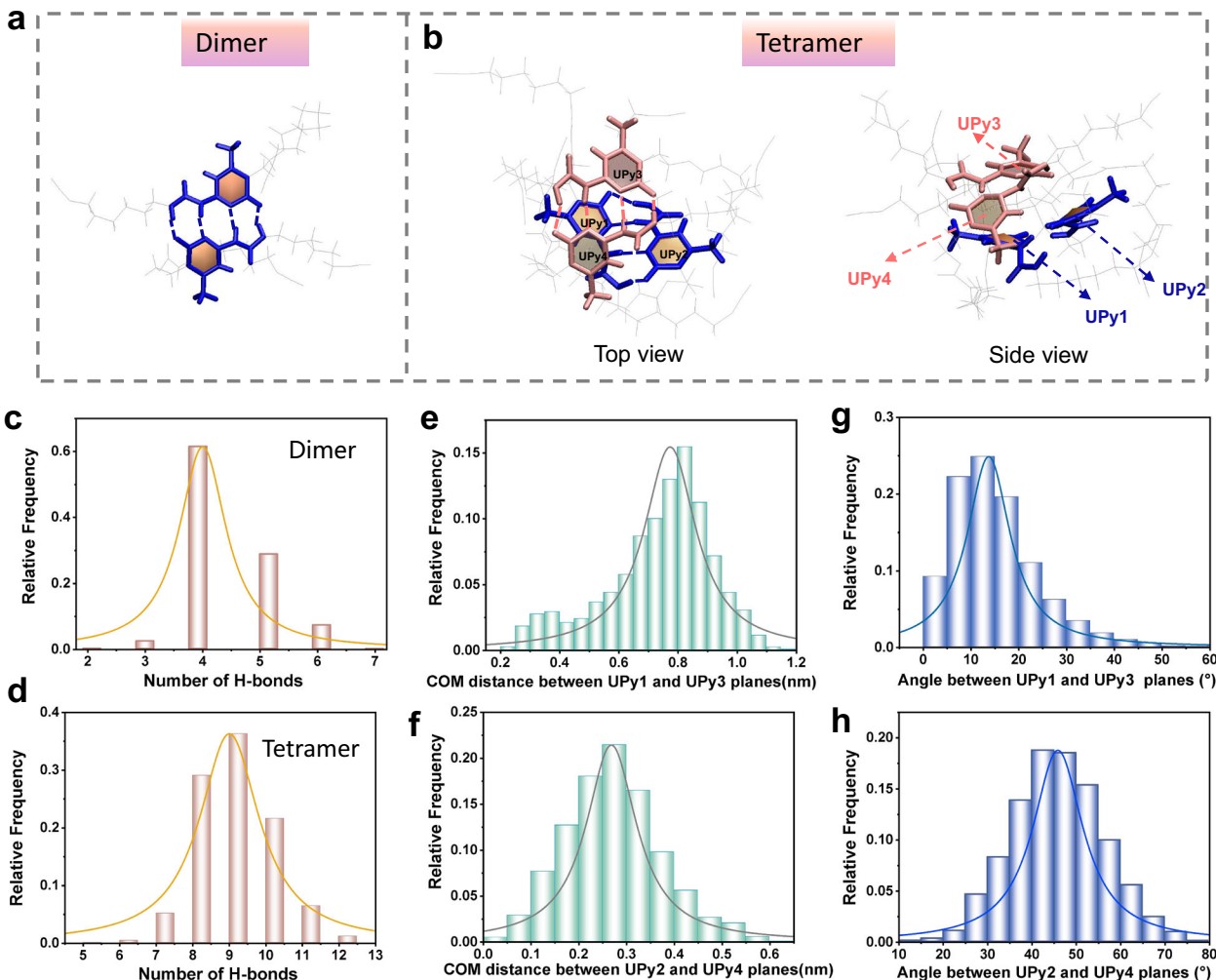

**Fig. 5 | Molecular dynamics simulations of UPy motifs. a**, **b** Spatial structure of UPy-DHDI dimers and tetramers in MD. **c**, **d** Statistics on the number of hydrogen bonds of UPy dimers tetramers. **e**, **f** Statistics of the interplanar COM distance of two UPy planes within tetramers. **g**, **h** Statistics of the interplanar angle of two UPy planes within tetramers.

properties. When spin-coated on glass, it causes only a 0.17% decrease in visible light transmittance (Supplementary Fig. 23a, b), making it suitable for applications requiring high light transmittance, such as surface coatings and optical device interfaces. It also shows good recyclability by hot-press molding (Fig. 6a) and solution casting. After being dissolved and remolded five times in DMF at 100 °C, the quantum yield remains stable (from 51.96% to 49.44%, Supplementary Figs. 25–27). The PLQY data from different batches also serves as evidence for the stability of the synthesis (Supplementary Fig. 24). At 40 °C, the fractured surface can heal efficiently within 10 min (Fig. 6b). The mild self-healing ability, high toughness, and reprocessability imparted by the dynamic hydrogen bonding network enhance the durability of the material under harsh conditions. Based on the reprocessability and self-healing properties of PU-UPy, it can be used as a fluorescent adhesive for restoring cultural relics. Under natural light, the cracks are concealed, not affecting the exhibition; under UV light, the cracks fluoresce, facilitating inspection and repair (Fig. 6c).

The fluorescence properties of the luminophore showing ESPT behavior are strongly correlated with hydrogen bond strength. These properties are influenced by solvent type and pH, demonstrating stimulus responsiveness[25,32]. Specifically, the fluorescence intensity of PU-UPy varies significantly between −40 and 120 °C (Fig. 6d, e), following a two-segment linear relationship. The inflection point corresponds to its melting temperatures (Supplementary Fig. 28), indicating

that UPy fluorescence can sensitively monitor polymer network mobility changes. Under a fluorescence microscope, these variations are visible to the naked eye within this temperature range. Additionally, the changes exhibit rapid response kinetics and a broad operational range, making this material ideal for use as a fluorescent temperature probe (Fig. 6f). Treatment with trifluoroacetic acid (TFA) caused significant fluorescence quenching of PU-UPy (quantum yield reduced to 2.16%, Supplementary Fig. 29), accompanied by a red shift of the emission peak from 410 nm (characteristic of UPy dimer) to 440 nm (attributed to CTE from the urethane moiety) (Fig. 6g). The emission spectra exhibited CTE features and excitation wavelength dependence (Supplementary Fig. 30), indicating disruption of the multiple-hydrogen-bond-confined luminescent centers. Upon partial removal of TFA by heating, the fluorescence intensity partially recovered, and the emission peak blue-shifted back to 410 nm (Fig. 6g), demonstrating its potential as an acidity probe. Fourier transform infrared (FTIR) and solid-state nuclear magnetic resonance (SSNMR) spectroscopy (Supplementary Figs. 31 and 32) confirmed that protons from TFA disrupted the multiple hydrogen-bonding network, thereby hindering the ESPT process and resulting in fluorescence quenching. This result further validates the ESPT-based luminescence mechanism of PU-UPy confined by multiple hydrogen bonds.

Similarly, FeCl₃ treatment induced significant fluorescence quenching (quantum yield <1%, Supplementary Fig. 33). NMR, FTIR

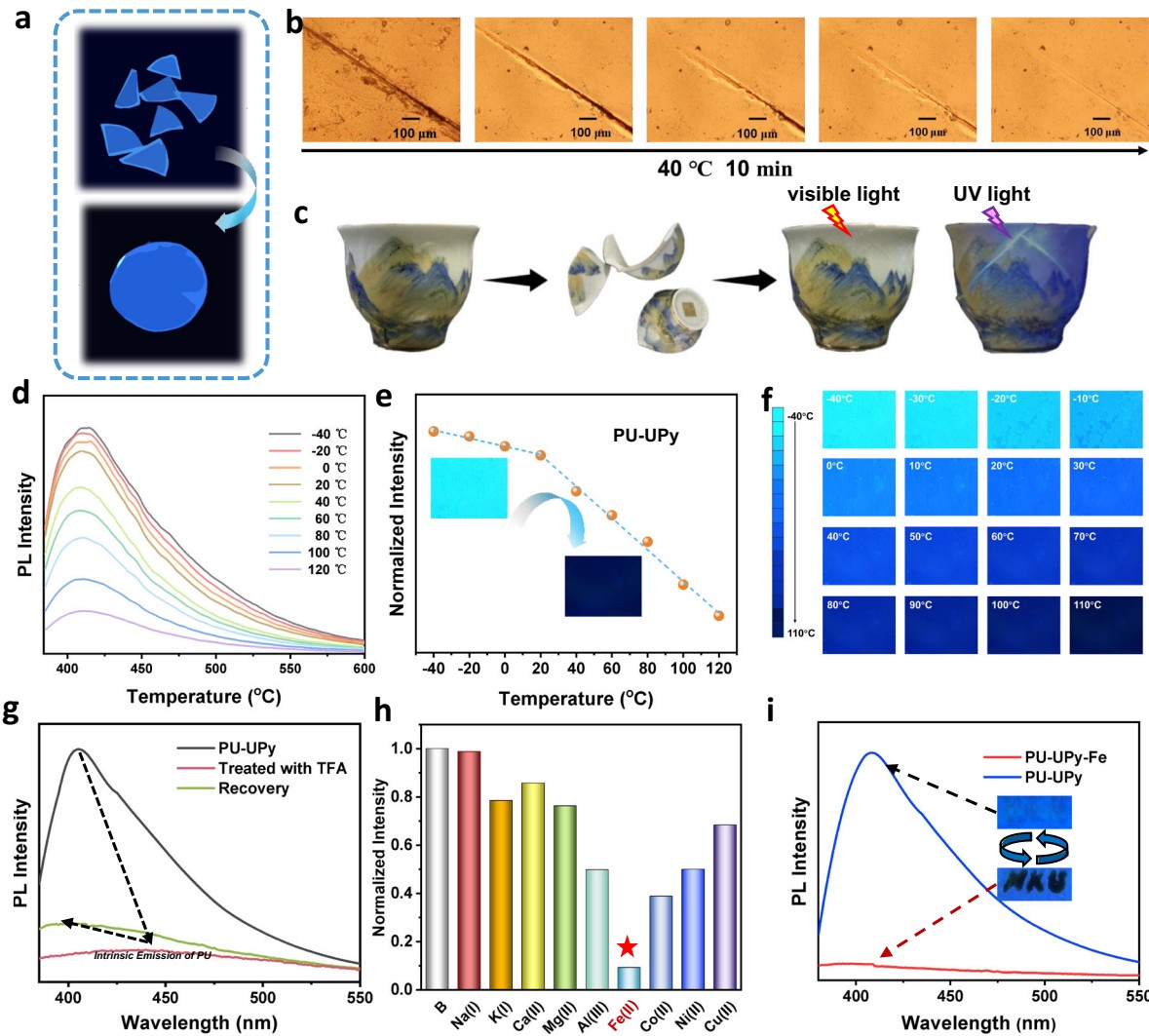

**Fig. 6 | Multiple applications of PU-UPy. a** Photo image of PU-UPy before and after recycling (under the condition of 365 nm UV lamp irradiation). **b** Optical microscopy images of PU-UPy before and after healing at 40 °C. **c** Photographs of ceramic cup repaired with the fluorescent adhesive (PU-UPy) under visible light and UV light. **d** PL spectra of PU-UPy at various temperature. **e** Plots of PL intensity versus temperature for PU-UPy under 365 nm UV light irradiation. **f** Fluorescent microscopy image of PU-UPy at various temperature. **g** PL spectra of PU-UPy before and after treated with TFA. **h** Relative PL intensities of PU-UPy with various metal ions. **i** PL spectra of PU-UPy before and after treated with FeCl₃ solution (0.001 M). Inset: photographs of PU-UPy before and after treated with FeCl₃ solution.

(Supplementary Figs. 31 and 32), and Raman spectroscopy (Supplementary Fig. 34) revealed that $Fe^{3+}$ introduction neither disrupted the multiple hydrogen-bonding structure nor induced new coordination. The formation of a new absorption band in the 400–600 nm range in the UV–vis spectrum (Supplementary Fig. 35), along with the broadening of the carbonyl peak in the solid-state $^{13}C$ NMR spectrum (Supplementary Fig. 36), indicates the presence of excited-state charge transfer between $Fe^{3+}$ and the carbonyl groups of UPy due to spatial proximity[47]. Leveraging this property, a 0.001 M FeCl₃ aqueous solution can be utilized as "ink" for writing on PU-UPy "paper" producing clearly visible text under 365 nm UV light (Fig. 6i). The quenched fluorescence is nearly fully restored upon ethanol wiping, demonstrating its rewritable capability and applicability in fluorescent anti-counterfeiting materials. DSC and SAXS analyses confirmed that FeCl₃ (and TFA) treatment did not alter the polymer's microphase-separated structure or thermal stability (Supplementary Fig. 37), thereby excluding structural changes as the quenching origin and highlighting the material's sensing stability. In a 20 μM PU-UPy DMF probe solution containing 200 μM of various ions (Na⁺, K⁺, Ca²⁺, Mg²⁺, Al³⁺, Fe³⁺, Co²⁺, Ni²⁺, Cu²⁺), only $Fe^{3+}$ induced substantial fluorescence

quenching to below 10% of its original PL intensity (Fig. 6h and Supplementary Fig. 38), exhibiting exceptional selectivity and validating its potential as a specific $Fe^{3+}$ probe. Collectively, the multifunctional stimulus-responsive behavior—integrating temperature monitoring, acidity sensing, specific detection of $Fe^{3+}$, and reversible anti-counterfeiting—demonstrates PU-UPy's capability as a versatile platform. We cultured L929 cells using an extract of PU-UPy and assessed its biocompatibility by measuring cell viability over a 1–5 days period. The results, presented in Supplementary Fig. 39, demonstrate that the extracts at various dilution ratios showed no cytotoxicity compared to the negative control, confirming the material's potential for applications in the biomedical field. The synergistic interplay between its multiple-hydrogen-bond-confined ESPT mechanism and controllable charge transfer underpins this material's foundation for developing intelligent fluorescent sensing systems.

In summary, inspired by the luminescent mechanism of GFP, this study proposes an approach to design and synthesize multifunctional fluorescent supramolecular polymers using dynamic quadruple hydrogen bonding of UPy motifs. By modulating the aggregated structure of UPy units, the ESPT process was activated,

achieving a quantum yield of up to 52%, thereby overcoming the limitations of low luminescent efficiency and single functionality in NTLPs. Experimental and theoretical calculations revealed that the configurational transformation of UPy dimers through the ESPT process in the excited state is the fundamental reason for the material's efficient luminescence. Additionally, SSNMR spectroscopy and time-resolved transient absorption spectroscopy further confirmed the critical role of quadruple hydrogen bonds of UPy in intermolecular proton transfer. The study also suggests that end groups with certain rigidity or polymer chains with hydrogen bonding interactions can improve the aggregated structure of UPy dimers, facilitate efficient ESPT, and suppress non-radiative transitions, thereby enhancing luminescent efficiency. The prepared fluorescent polymer material not only exhibits excellent luminescent properties but also integrates multiple functionalities such as high toughness, self-healing, reprocessability, and stimulus responsiveness. These multifunctional characteristics make the material promising for applications in fluorescent coatings, fluorescent adhesives, biofluorescence, and multi-sensing. This study not only provides a paradigm for the molecular design of NTLPs but also lays the foundation for developing a multifunctional material platform integrating optics, mechanics, and intelligent responses, which is of great significance for the development of next-generation biointegrated devices and adaptive optoelectronics.

## Methods

### Chemical and materials

A list of all chemicals and materials can be found in the supplementary information section.

### Synthesis and sample preparation

The synthesis procedure of UPy precursor was shown in Supplementary Fig. 1. A mixture of 2-acetylbutyrolactone ($5.12 \times g$, 40 mmol) and guanidine carbonate ($3.6 \times g$, 40 mmol) was refluxed in absolute ethanol (40 mL) in the presence of $Et_3N$ (11.0 mL, 80 mmol) for 1 h. Then the mixture became clear, and the reaction continued for 3 h. The final obtained product was precipitated and filtered to yield a light-yellow solid, which was further washed with ethanol for three times and dried in vacuum to obtain a white powder solid, i.e., 5-(2-hydroxyethyl)-6-methyl-2-aminouracil. The chemical structure was confirmed by [1]H-NMR (Supplementary Fig. 2).

The synthesis procedure of UPy-DHDI was also shown in Fig. 1. The 5-(2-hydroxyethyl)-6-methyl-2-aminouracil ($0.85 \times g$, 5.0 mmol) was dissolved in a mixture of 10 mL of 1,6-diisocyanatohexane (HDI) and 1 mL of pyridine under a nitrogen atmosphere. The mixture solution was stirred at 100 °C for 1.5 h followed by precipitation into tenfold excess of petroleum ether. The resulting solid was then washed with pentane and dried in vacuum to yield a white powder. The chemical structure of UPy-DHDI was confirmed by [1]HNMR spectra as shown in Supplementary Fig. 3.

The synthesis procedure of PU-UPy was also shown in Supplementary Fig. 4. PTMG-2000 ($2.0 \times g$, 1 mmol) firstly reacted with UPy-DHDI ($1.010 \times g$, 2 mmol) in anhydrous N,N-Dimethylformamide (DMF, 15 ml) at 70 °C under nitrogen atmosphere for 1 h, yielding a pre-polymer. Subsequently, BDO (1,4-butanediol, $0.09 \times g$, 1 mmol) and the catalyst dibutyltin dilaurate (0.5 mol% for isocyanate or alcohol units) in anhydrous N,N-Dimethylformamide (DMF, 2 ml) were added into the mixture and further stirred for another 12 h at 70 °C. The solution was poured into a Teflon casting dish dried under 70 °C overnight followed by 70 °C for 48 h under vacuum. After drying, the resulting polymer sheet was peeled off from the Teflon mold and used for further measurements. The chemical structure was confirmed by [1]H-NMR (Supplementary Fig. 5) and FTIR (Supplementary Fig. 6). Blank sample PU-b was synthesized using the same synthesis method (UPy-DHDI was replaced with HDI).

## Characterization

Solution NMR experiments were performed on a Bruker AVANCE III NMR spectrometer with a proton resonance frequency of 400.13 MHz. The samples were dissolved in deuterated chloroform or DMSO-d6 or DMF-d with a small amount of TMS as the internal reference standard. Solid-state NMR experiments were performed on a Bruker AVANCE NEO wide-bore (89 mm) NMR spectrometer operating at a proton frequency of 400.18 MHz. A conventional 1.3 mm double resonance MASDVT400W1 probe was used, the [1]H 90° pulse length was 1.5 μs and the recycle delay was set as 6 s. The fast magic angle spinning (MAS) was automatically controlled at 60 kHz within ± 1 Hz with a MAS speed controller. For [1]H two-dimensional (2D) double-quantum/single-quantum (DQ/SQ) NMR experiments, the BABA-xy16 pulse sequence[48] was used for DQ coherence excitation and reconversion. The spectral width on t1 and t2 dimension was set as 60 kHz and 25 kHz, respectively.

The FT-IR spectra were obtained using a TENSOR II(Bruker, Germany). The infrared spectra were recorded with a resolution of 8 $cm^{-1}$ and 16 scans per sample.

UV–vis absorption of solutions was recorded on the UV–vis spectrophotometer (Specord plus 210). UV–vis absorption of solid were recorded on the UV–vis spectrophotometer equipped with integrating spheres (Hitachi U-4100).

The steady-state excitation, emission spectra, and quantum yield of solid and solution samples were measured by steady-state transient fluorescence spectroscopy (Edinburgh FS5).

The fs-TAS spectra were obtained with a self-built femtosecond transient absorption spectroscopy. The light source we used is a Pharos laser with a center wavelength of 1030 nm, an average power of 20 W, a pulse adjustable range of <290 fs-10 ps, and a repetition rate of 1 0 kHz−200 kHz. The optical parametric amplifier is used to generate pump pulses with specific wavelengths, and the optical pulse width of the compressed signal is <100 fs. We adjust the arrival time of pump-probe light through optical delay line, in which the stepping accuracy of optical delay line is 0.1 μm, corresponding to the time of light passing time is 0.33 fs, and can provide a time window of 8 ns. The specific optical path was built with reference to the previous work[49]. The transient absorption optical path diagram is shown in Supplementary Figs. 12 and 13.

DSC measurements were performed on a Mettler-Toledo DSC1 differential scanning calorimeter with a heating rate of 10 K/min under nitrogen atmosphere. About 10 mg samples were encapsulated in 40 μL aluminum pans before measurements.

XRD measurement was carried out on a Bruker Model D8 FOCUS X-ray diffractometer with Cu Kα radiation (λ= 1.5406 Å) at a generator voltage of 40 kV and a current of 40 mA.

The SAXS experiments were performed at room temperature using a Xenocs Xeuss3.0 SAXS system operated at 50 kV and 30 mA. The wavelength of the incident X-ray beam from Cu Kα radiation was 0.154 nm. The long period d is inversely related to the wave vector at the scattering peak, $d = 2/q$.

TGA experiments were conducted on a NETZSCH TG 209 instrument at a linear heating rate of 10 °C $min^{-1}$ from 25 to 600 °C under nitrogen atmosphere.

Mechanical properties of the samples were tested using a UTM6103 microcomputer-controlled desktop electronic universal testing machine. Micrographs of the self-healing experiments were captured using an Olympus BX53 microscope equipped with a hot stage.

TD-DFT calculations were performed using the Gaussian 16 software. Structural optimization was conducted at the B3LYP/6-31G* level of theory, utilizing the Gaussian 16 and Multiwfn software packages[40,41].

To obtain the equilibrium conformations of UPy-DHDI dimers and tetramers, we performed molecular dynamics (MD) simulations on UPy-DHDI in the poor solvent hexane. Atomic types and force field

parameters for UPy-DHDI and hexane were assigned using the General Amber Force Field[50] (GAFF). The atomic partial charges of each atom in UPy-DHDI were generated by the restrained electrostatic potential (RESP) fit method based on the calculated electrostatic potential at the M06-2×/6-31G** level by the Gaussian 16 package[51–53]. The initial configurations of the UPy-DHDI dimers and tetramers were generated through randomly placing one UPy-DHDI dimer (two molecules) and two parallel UPy-DHDI dimers (four molecules) into a 6 nm cubic box solvated by hexane, respectively. For each system, the energy minimization was first performed using the steepest descent algorithm. Subsequently, 2 ns pre-equilibrated MD simulations were performed under the NVT (T = 300 K) and NPT ensembles, respectively. Finally, a 100 ns production MD simulation was carried out for each system under the NPT ensemble (T = 300 K, P = 1 bar) to obtain equilibrated dimers and tetramers. During the equilibrium simulation, temperature and pressure were controlled by the velocity rescaling thermostat[54] and Berendsen barostat[55], respectively. The classical Newtonian equations of motion were performed by leapfrog algorithm with an integral step of 2 fs. For all systems, the conformations of the production MD simulations were stored every 4 ps for analysis. Electrostatic interactions were treated using the particle mesh Ewald (PME) method.[56,57] All MD simulations were performed using the GROMACS package (version 5.1.5)[58]. Trajectory analysis was performed with the help of the utility tools included in the GROMACS package[58] and VMD software[59].

The biocompatibility of PU-UPy was evaluated using L929 mouse fibroblasts. A material extract was prepared at 0.1 × g/mL in serum-free DMEM and incubated at 37 °C for 24 h. The extract was diluted with complete medium to various concentrations. L929 cells were seeded in 24-well plates and, after 24 h, exposed to the diluted extracts, complete medium (control), or blank medium. After 1, 3, and 5 days, cell viability was measured with a CCK-8 assay. The absorbance at 450 nm was recorded, and relative viability was calculated. All groups were tested in sextuplicate. The error bars in the figures represent the standard deviation derived from three independent replicate measurements for each data point. They were calculated as the positive square root of the sample variance for each set of triplicates and are displayed as ±1 SD (More details about the cell experiments are listed in the supplementary information).

## Data availability
The source data generated in this study, including spectroscopic, computational, and mechanical data, have been deposited in the Figshare database under accession code (https://doi.org/10.6084/m9.figshare.30574364). All other data supporting the findings of this study are available in the article and its Supplementary files.

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

## Acknowledgements

This work was supported by the National Natural Science Foundation of China (22003028, awarded to F. W and 22073051, granted to P. S).

## Author contributions

H. Zuo and F. Wang conceived and initiated the study. H. Zuo conceptualized, prepared samples, performed characterization experiment and analysis, visualized data, and prepared figures. Y. Zeng and X. Zheng performed molecular dynamics simulation and analysis. Q. Gao, Z. Xiang, and C. Jia set up the ultrafast spectroscopy device, performed experiment and data analysis. Q. Zhang and Y. Zhu performed DFT simulation and data analysis. P. Sun supported experiments. B. Z. Tang and F. Wang supervised the study, and designed experiments and analyses. H. Zuo and F. Wang wrote the paper with P. Sun and B. Z. Tang. All authors read and approved the manuscript.

## Competing interests

The authors declare no competing interests.
