## [Transparent Peer Review file · Nature Communications]

Non-traditional Fluorescence in Quadruple Hydrogen Bonded Supramolecular Polymers

Corresponding Author: Professor Fenfen Wang

Version 0:

Reviewer comments:

Reviewer #1

(Remarks to the Author)

This paper presents the advancement of non-traditional luminescent polymers (NTPs) by developing a multifunctional fluorescent supramolecular polyurethane (PU-UPy) inspired by green fluorescent protein (GFP). The authors achieve a high photoluminescence quantum yield (PLQY) of up to 52% through the modulation of ureidopyrimidinone (UPy) units, leveraging excited-state proton transfer (ESPT) and quadruple hydrogen bonding. The study investigates the luminescence mechanism and validate its multifunctionality. However, according to a high standard for Nature Communications, several shortcomings and issues need to be addressed.

1. Authors claim "Non-traditional luminescent polymers (NTPs) exhibit remarkable advantages in bio-diagnostics and intelligent materials", but in this paper any information about bio-diagnostics and intelligent materials did not emerge. Some bio-diagnostics experiments should be supplemented to improve the quality of the paper and self-consistent their claim.

2. The authors claim that their work presents the "first proposal of its ESPT luminescence mechanism" for UPy motifs. While this may be true for ESPT, the paper does mention a prior related reference for "isomerization behavior of UPy". More precise explanations how they differ from previously reported photophysical behaviors of UPy would be necessary. Especially the references about the luminescence works using UPy are missed.

3. The paper proposes that the UPy dimer undergoes "cooperative multi-proton transfer and dynamic hydrogen bond reorganization" upon excitation. While the fs-TA spectroscopy provides compelling evidence for the keto \rightarrow enol tautomerization, the paper does not provide detailed evidence that the transfer is "cooperative" or "multi-proton". While this is a reasonable assumption given the quadruple hydrogen bonds, a insightful analysis or discussion of this specific kinetic pathway, perhaps with computational modeling of the transition state, would strengthen this claim.

3. Authors demonstrate that the PLQY remains stable after five cycles of dissolution and remolding. However, a key problem is the lack of a statement on the reproducibility of the initial 52% PLQY across multiple synthesis batches. Providing data on batch-to-batch consistency would add confidence in this key claim and support the robustness of the synthesis.

4. Authors claim that the fluorescence quenching by FeCl_3 to excited-state intermolecular charge transfer. While this is a valid hypothesis, the authors also state that FeCl_3 treatment did not disrupt the multiple hydrogen-bonding structure. The work does not provide a detailed mechanism for how the Fe^{3+} ion interacts with the UPy motif or other parts of the polymer to facilitate charge transfer without affecting the crucial hydrogen bond network. A more detailed exploration of this interaction would be added.

5. The work describes UPy-DHI as having "extremely weak fluorescence emission" with a quantum yield "below 10%". However, the weak fluorescence is then attributed to the "residual UPy dimer with quadruple hydrogen bonds", which is a potential contradiction. The authors should clarify whether the weak emission is from the bulk crystalline material or a minor amorphous component. Quantifying the amount of this "residual UPy dimer" and its contribution to the overall fluorescence would provide a more precise and academically rigorous explanation.

6. The authors claim that the large interplanar spacing in UPy-DHDI indicates a "non- π - π close packing arrangement," for luminescence. The term itself is somewhat vague, and a clearer explanation of how this specific type of packing, as opposed to close packing in the crystalline UPy-DHI, prevents non-radiative decay pathways (such as aggregation-caused quenching, ACQ) and facilitates efficient ESPT would improve the clarity and rigor of the argument.

7. The disordered amorphous structure of UPy-DHI is responsible for the high luminescence, while the crystalline structure of the control sample UPy-DHI leads to weak emission. The authors attribute this to the amorphous structure suppressing exciton annihilation. However, the paper does not provide a detailed mechanistic explanation of why the amorphous state is more favorable for the ESPT process itself. A insightful evidence of how the dynamics of the hydrogen bond network or the conformational freedom in the amorphous state specifically facilitating proton transfer would be offered.

8. The paper posits the existence of a second enol form (Enol₂) to explain the long-wave emission peak observed in CHCl₃ solution. While this is supported by theoretical calculations that show a good match between the predicted and experimental wavelengths, this evidence is primarily computational. A significant flaw is the absence of any direct experimental verification of the Enol₂ form's existence, which would substantially strengthen the proposed ESPT mechanism. Further experimental techniques, such as time-resolved spectroscopy specifically designed to probe for this second tautomer, could be necessary.

Reviewer #2

(Remarks to the Author)
See the attached file.

Version 1:

Reviewer comments:

Reviewer #1

(Remarks to the Author)
The authors have resolved my concern issues, I think this work can be polished

Reviewer #2

(Remarks to the Author)
Zuo and co-authors did a great work considering the comments and questions of both reviewers. Especially, they completed the experimental part going upon the test of biocompatibility of their systems, demonstrating they are non cytotoxic and that they can be used in biomedical applications but they also added time-resolved emission spectroscopy to prove the existence of a particular enol form. The information they added, for example about the energy barriers for ESPT, support the affirmations within the manuscript. The clarifications about some technical details regarding the calculations will also help the future readers and finally all the questions were addressed and the missing information added to the manuscript.

made.

The manuscript by Zuo *et al* deals with unexpected fluorescence properties induced by a well know moiety (ureidopyrimidinone, UPy) within a polymer. A combination of experimental (synthesis, spectroscopies, various applications) and theoretical (TD-DFT, MD) approaches has been convincingly provided to present, rationalize and prove the efficiency and the potential of such systems. Very interestingly, they propose a mechanism involving tautomerization and excited state proton transfer to explain the boost in fluorescence efficiency while the restriction of motions is often argued to explain it. Please find below some questions and comments I had after reading the manuscript. Some of them should be taken into account to clarify or precise some points within the main text.

Comments & questions:

- As no details about the TD-DFT calculations are provided (oscillator strength, nature and description in terms of orbitals/involved states), does the maximum absorption wavelength corresponds to S_1 state, as only the S_1 was optimized to retrieve the emission wavelength? Are HOMO and LUMO the only orbitals involved within the transition?
- How was the curve of figure 9b in SI obtained? Was it a simple scan (relaxed or not?) with potential energies reported or is it a “real” energy barrier with frequency calculations involved? This should be clarified within the manuscript and SI.
- Does the tautomerization process take place at the ground state (GS), towards the enol form, or only in the excited state (ES)? Is there a competition between the tautomerization at the GS and the ES? In the excited state, following figure 9b in SI, the keto form seems to be more stable than the enol form, is it compatible with the following steps?
- Are the 4 H-bonds always effective during MD simulations? What are the extra H-bonds observed? Do they correspond to “peripheric” H-bonds? It should be stated that the keto form was considered for MD simulation. Was the MD simulation of the keto form done with the GS or ES structure (if the structures/charges/density repartition are different between GS and ES)? Why the enol1 and enol2 were not considered during MD simulations?

- Why are there such harsh thresholds on the emission spectra of UPy-DHI (figure 7b in SI)? Were the spectra recorded over the entire wavelength range?
- Why was hexane used for MD simulations and not DMF or chloroform?
- On figure 2a, the second ESPT appears as reversible (double arrow) but not the first one (simple arrow). Why? Is it due to energetic considerations? This can be added to the figure and/or the text if the authors have the energy barriers or an approximation of it.
- Are the Enol2 dimers also stable as there are only 2 intermolecular H-Bonds, compared to 4 H-Bonds for the keto/enol1 dimers?

Overall, the manuscript is dense in results, the explanations are convincing, and it provides new results about a multifunctional fluorescent polymer, and I recommend its publication after minor corrections/precisions (see above).

Typos:

Figure 5: (e) Plots of PL intensity versus **temperature (?)** for PU-UPy under 365 nm UV light irradiation.

Spectral characterization revealed that UPy-DHI exhibited a quantum yield below 10% and displayed excitation wavelength-dependent fluorescence emission (Supplementary Figs. 10b, d). → **There is no Fig 10b, d in SI.**

Figure 12 in SI. Graph b does not represent the emission spectra at different wavelength but at different concentration.

Figure 9b in SI. Unit of the y axis should be kcal mol⁻¹

Response Letter to Reviewers

Reviewer #1 (Remarks to the Author):

This paper presents the advancement of non-traditional luminescent polymers (NTLPs) by developing a multifunctional fluorescent supramolecular polyurethane (PU-UPy) inspired by green fluorescent protein (GFP). The authors achieve a high photoluminescence quantum yield (PLQY) of up to 52% through the modulation of ureidopyrimidinone (UPy) units, leveraging excited-state proton transfer (ESPT) and quadruple hydrogen bonding. The study investigates the luminescence mechanism and validates its multifunctionality. However, according to a high standard for Nature Communications, several shortcomings and issues need to be addressed.

Reply: We sincerely appreciate the reviewer's comments and insightful suggestions. In response, we have carefully revised the manuscript (with changes highlighted in yellow) and provide a point-by-point response to all comments below.

1. Authors claim “Non-traditional luminescent polymers (NTLPs) exhibit remarkable advantages in bio-diagnostics and intelligent materials”, but in this paper any information about bio-diagnostics and intelligent materials did not emerge. Some bio-diagnostics experiments should be supplemented to improve the quality of the paper and self-consistent their claim.

Reply: We thank the reviewer for pointing this out. In the introduction, we have outlined the applications of various NTLPs, and included the statement “Non-traditional luminescent polymers (NTLPs) exhibit remarkable advantages in bio-diagnostics and intelligent materials”. However, given that each material possesses certain advantages and limitations, the descriptions provided are not intended to imply that every NTLP is suitable for biomedical diagnostics or/and smart materials. The core objective of this paper is to elucidate a novel luminescence mechanism and discovery of a new luminescent polymer with breakthrough in the versatile integration of functionalities within NTLP materials. The reported material demonstrates greater suitability for macro-scale smart materials, as exemplified by its applications in temperature response and ion detection, among others. Nevertheless, we fully acknowledge the reviewer’s perspective that further exploration and expansion of the material’s potential in biomedical diagnostics are warranted. To investigate the biocompatibility of the material, we co-cultured the extract of PU-UPy with L929 cells and assessed its biocompatibility by measuring cell viability. The corresponding data have been added in the

revised Supplementary Fig. 37. The extracts at various dilution ratios-maintained cell viability above 90% over culture periods of 1, 3, and 5 days, showing no significant difference compared to the negative control group. The results demonstrate that PU-UPy exhibits excellent biocompatibility and holds potential for applications in bioimaging and diagnostics. However, limited by the deficiencies in our experimental conditions for biosensing and other aspects, further relevant experiments may be the direction of our future research.

We have added a discussion of the biocompatibility results to the revised manuscript, as highlighted: *“We cultured L929 cells using an extract of PU-UPy and assessed its biocompatibility by measuring cell viability over a 1–5 days period. The results, presented in Supplementary Figure. 37, demonstrate that the extracts at various dilution ratios showed no cytotoxicity compared to the negative control, confirming the material’s potential for applications in the biomedical field”*.

Figure R1. Cell viability of different dilution factors of the experimental materials group (compared with the negative control group).

2. The authors claim that their work presents the "first proposal of its ESPT luminescence mechanism" for UPy motifs. While this may be true for ESPT, the paper does mention a prior related reference for "isomerization behavior of UPy". More precise explanations how they differ from previously reported photophysical behaviors of UPy would be necessary. Especially the references about the luminescence works using UPy are missed.

Reply: We sincerely appreciate the reviewer's valuable insight and suggestion. The "UPy isomerization" mentioned in the cited reference refers to a ground-state conformational change achieved through chemical modification or physical treatment, which does not involve photophysical processes. This fundamentally differs from the photoinduced excited-state proton transfer (ESPT) process reported in our work. In accordance with the reviewer's suggestion, we have

provided a more detailed explanation in the revised manuscript.

In the introduction of this paper, we present a comprehensive discussion on the luminescence research related to UPy, particularly highlighting “UPy has been employed as an auxiliary unit to enhance traditional fluorophores²⁹, or as an energy transfer medium to improve phosphorescence efficiency”. In these previous studies, UPy was not utilized as a luminophore but rather served as an auxiliary unit to construct supramolecular assemblies. To the best of our knowledge, no literature has reported the intrinsic fluorescence behavior of UPy. This work represents the first report of this phenomenon, which also constitutes the key innovation of our study. We have expressed the discussion on the photoinduced isomerization behavior of UPy more clearly in the revised manuscript, as highlighted: “However, its intrinsic luminescent properties and photoinduced keto – enol tautomerism have remained undiscovered until now ”, and “It has been reported in the literature that the existence of the inherent tautomerization of heterocyclic ring-based system such as UPy derivatives³⁸. And the structure control of intramolecular proton tautomerism has been an inspired and fruitful research area in supramolecular chemistry³⁰. In fact, the equilibrium of proton tautomer of UPy is solvent and temperature dependent⁴⁰, referring to a permanent conformational change in the ground state. Based on the experimental results presented above as well as the existence of the inherent tautomerization of UPy, we hypothesize that UPy undergo photoinduced tautomerism in the excited state for the first time”.

3. The paper proposes that the UPy dimer undergoes "cooperative multi-proton transfer and dynamic hydrogen bond reorganization" upon excitation. While the fs-TA spectroscopy provides compelling evidence for the keto → enol tautomerization, the paper does not provide detailed evidence that the transfer is "cooperative" or "multi-proton". While this is a reasonable assumption given the quadruple hydrogen bonds, a insightful analysis or discussion of this specific kinetic pathway, perhaps with computational modeling of the transition state, would strengthen this claim.

Reply: We sincerely thank the reviewer for raising this critical question, which prompted us to perform more detailed calculations and thereby refine our mechanistic hypothesis. To describe the proton transfer process more clearly, we carried out energy calculations for the intramolecular proton transfer and intermolecular proton transfer processes step by step, the computational results were shown in Fig. R2. It was found that directly completing the intermolecular proton transfer

(Proton from N3 – C4) from the keto configuration to form the enol structure requires overcoming a relatively high energy barrier (2.97 eV), which is unlikely to occur under conventional photoexcitation. However, we observed that the intramolecular ESPT (N1 – C2) process only requires an energy barrier of 1.12 eV, suggesting that the intramolecular ESPT is likely to occur preferentially. Furthermore, after the proton transfers from N1 to C2, the electronic distribution of the molecule changes, leading to an increased charge density around the N3 atom, which facilitates proton transfer between N3 and C4. Based on these two-step calculation results, we propose that due to the lower energy barrier, intramolecular proton transfer occurs prior to intermolecular proton transfer. The intramolecular proton transfer provides a favorable configuration for the subsequent intermolecular proton transfer, reducing the energy barrier and thereby promoting the overall process. These additional computational results provide supporting evidence for the hypothesized concerted proton transfer mechanism. We have expanded the discussion regarding these new findings as highlighted: *“To more accurately describe the proton transfer process, we performed potential energy surface scans for both the intramolecular and intermolecular proton transfer pathways (Supplementary Fig.13). The results suggested that direct intermolecular proton transfer from the keto configuration faces a high energy barrier (2.97 eV), making it unlikely under conventional photoexcitation. In contrast, the intramolecular pathway has a much lower barrier (1.12 eV), suggesting it occurs preferentially. Once the intramolecular transfer is complete, the resulting electronic rearrangement significantly reduces the barrier for subsequent intermolecular transfer, facilitating a concerted mechanism. Therefore, the ESPT of UPy dimers should be a intramolecular-intermolecular cooperative multi-proton transfer process”* in the revised manuscript.

Figure R2. Potential energy surface scanning for intramolecular (a) intermolecular (b) ESPT.

4. Authors demonstrate that the PLQY remains stable after five cycles of dissolution and remolding. However, a key problem is the lack of a statement on the reproducibility of the initial 52% PLQY across multiple synthesis batches. Providing data on batch-to-batch consistency would add confidence in this key claim and support the robustness of the synthesis.

Reply: We sincerely appreciate the reviewer's valuable suggestion. As requested, we tested the photoluminescence quantum yield (PLQY) of three different batches of the material, with each batch measured at three different points and averaged. The average PLQY values for the three batches were 51.96%, 50.90%, and 52.57%, respectively. The result calculated based on standard deviation is 51.81 ± 0.7 . These results confirm the reproducible high quantum yield and the stability of the material synthesis. We have presented the PLQY spectra of three different batches in Fig. R3. We have emphasized this point in the revised manuscript as highlighted: “*The PLQY data from different batches also serves as evidence for the stability of the synthesis (Supplementary Fig. 22)*”.

Figure R3. PLQY testing of three different batches of PU-UPy (a-c).

5. Authors claim that the fluorescence quenching by FeCl_3 to excited-state intermolecular charge transfer. While this is a valid hypothesis, the authors also state that FeCl_3 treatment did not disrupt the multiple hydrogen-bonding structure. The work does not provide a detailed mechanism for how the Fe^{3+} ion interacts with the UPy motif or other parts of the polymer to facilitate charge transfer without affecting the crucial hydrogen bond network. A more detailed exploration of this interaction would be added.

Reply: We thank the reviewer for this suggestion. To further investigate the interaction mechanism between Fe^{3+} and the material, we conducted additional UV-vis absorption (Fig. R4) and ^{13}C CPMAS solid-state nuclear magnetic resonance (SSNMR) experiments (Fig. R5). The relevant results have been included in Supplementary Figs. 33-34. The UV-vis absorption spectra revealed that after binding with FeCl_3 , a new absorption band emerged in the 400–600 nm range for PU-UPy, which differs from the intrinsic $n\text{-}\pi^*$ absorption band of UPy or the d-d transition band of Fe^{3+} . This provides direct evidence for the formation of a charge-transfer complex. Fig. R5 shows the expanded UPy region of the ^{13}C CPMAS NMR spectra of both PU-UPy and PU-UPy-Fe samples. Upon the addition of Fe^{3+} , significant line broadening (measured at 25% peak height) was observed for the C=O peak at 172.0 ppm and the peak at 114 ppm in UPy. Moreover, all UPy peaks shifted toward lower frequency (increased shielding), indicating an interaction between Fe^{3+} and the carbonyl groups of UPy. These results further clarify the site where charge transfer occurs. In Supplementary Fig. 30 (Fig. R6) of the original manuscript, we presented the ^1H MAS spectrum of the Fe^{3+} -treated sample. The peaks associated with multiple hydrogen bonding remained unchanged in position and intensity, suggesting that the hydrogen-bonding configuration remained relatively stable. Additionally, Raman spectroscopy results indicated no significant formation of new

coordination bonds, implying that the interaction between Fe^{3+} and UPy is not a strong chemical binding but rather results from spatial proximity and electron cloud overlap. Collectively, these findings elucidate the specific interaction mode between Fe^{3+} and UPy and the underlying quenching mechanism. This point has been emphasized in the revised manuscript. The added text (highlighted for review) states: “The formation of a new absorption band in the 400–600 nm range in the UV-vis spectrum (Supplementary Fig. 33), along with the broadening of the carbonyl peak in the solid-state ^{13}C NMR spectrum (Supplementary Fig. 34), indicates the presence of excited-state charge transfer between Fe^{3+} and the carbonyl groups of UPy due to spatial proximity”.

Figure R4. Normalized UV-vis absorption spectra of PU-UPy, PU-UPy+FeCl₃ and FeCl₃ in DMF.

Figure R5. Solid-state ^{13}C CPMAS NMR spectrum of PU-UPy and PU-UPy-Fe.

Figure R6. ¹H-Solid state NMR spectra of PU-UPy, PU-UPy-TFA and PU-UPy-Fe.

6. The work describes UPy-DHI as having "extremely weak fluorescence emission" with a quantum yield "below 10%". However, the weak fluorescence is then attributed to the "residual UPy dimer with quadruple hydrogen bonds", which is a potential contradiction. The authors should clarify whether the weak emission is from the bulk crystalline material or a minor amorphous component. Quantifying the amount of this "residual UPy dimer" and its contribution to the overall fluorescence would provide a more precise and academically rigorous explanation.

Reply: We thank the reviewer for their valuable comments and insightful suggestion. To investigate the effect of amorphous regions on the weak fluorescence emission, we designed the following experiment: A chloroform solution of UPy-DHI was quenched in liquid nitrogen and then rapidly freeze-dried to remove residual chloroform, thereby obtaining a UPy-DHI sample with poor crystallinity. XRD results (Fig. R7) confirmed that this quenching treatment significantly reduced the crystallinity and substantially increased the amorphous content of the sample. Quantum yield testing (Fig. R8) was employed to evaluate the changes in the fluorescence properties of the samples before and after the quenching process. ¹H DQ/SQ solid-state NMR experiments (Fig. R9) were conducted to examine changes in the hydrogen-bonding configurations before and after the quenching process. We observed that although the amorphous content increased notably, no significant change in the quantum yield (6.4% – 6.8%) for quenched UPy-DHI sample. This indicates that the fluorescence does not come from the amorphous components in the samples. In addition, the ¹H DQ/SQ NMR results indicate that the hydrogen-bonded structures of the UPy-DHI

samples remain largely unchanged after liquid nitrogen treatment, with a significant portion of UPy groups not forming quadruple hydrogen bonds. This suggests that the quadruple hydrogen-bond network in UPy-DHI is disrupted compared to that in UPy-DHDI, thereby preventing the establishment of an effective ESPT pathway. However, owing to the limited resolution of the ^1H solid-state NMR spectra, a quantitative analysis of the hydrogen-bonded UPy groups is unfeasible. Nonetheless, the reviewer's question is undoubtedly valuable and will be pursued in our future studies. Based on these new results, we have provided a more detailed explanation of this issue in the revised manuscript. The added text as highlighted states: “*Subjecting the UPy-DHI sample to liquid nitrogen quenching produced an amorphous material (Supplementary Fig. 14) that still exhibited weak fluorescence (Supplementary Fig. 15), confirming that crystallization is not the origin of the fluorescence differences*”, and “*Notably, the ^1H NMR spectrum of UPy-DHI—both crystalline and amorphous—shows a distinct signal for the isolated proton Hc' , which is not involved in UPy hydrogen bonding (Fig. 3e and Supplementary Fig. 18). This directly demonstrates the disruption of the quadruple hydrogen-bonding network in UPy-DHI, which we propose suppresses the intermolecular proton transfer process, leading to its markedly weaker fluorescence*”.

Figure R7. XRD patterns of UPy-DHI before and after quenching treatment.

Figure R8. PLQY testing of UPy-DHI before and after quenching treatment

Figure R9. Solid-state NMR two-dimensional DQ/SQ correlation spectra of UPy-DHI sample before and after quenched in liquid nitrogen.

7. The authors claim that the large interplanar spacing in UPy-DHDI indicates a "non- π - π close packing arrangement," for luminescence. The term itself is somewhat vague, and a clearer explanation of how this specific type of packing, as opposed to close packing in the crystalline UPy-DHI, prevents non-radiative decay pathways (such as aggregation-caused quenching, ACQ) and facilitates efficient ESPT would improve the clarity and rigor of the argument.

Reply: We sincerely appreciate the reviewer's suggestion. For clarity, the term "non-compact packing arrangement" has now been changed and more specifically described as "larger molecular spacing and more disordered molecular orientation." We have accordingly revised the description of how this packing type influences the luminescence mechanism. The "larger molecular spacing and more disordered molecular orientation" directly reduce intermolecular π - π stacking. This promotes efficient fluorescence generation through two mechanisms. On one hand, π - π stacking structures can act as quenching sites for excitons, and this specific packing configuration reduces such quenching sites, ensuring the efficient occurrence of the ESPT process. On the other hand, the weakening of π - π stacking suppresses the ACQ (aggregation-caused quenching) effect, preventing the fluorescence generated via the ESPT process from dissipating through non-radiative transitions. We thank the reviewer for this valuable suggestion, which guided us in more clearly describing the influence of the aggregated state structure on the fluorescence mechanism. We have modified the relevant descriptions in the revised manuscript as highlighted: *"The increased molecular spacing and more disordered molecular orientation directly weaken intermolecular π - π stacking. This structural change enhances fluorescence efficiency through two distinct mechanisms. First, by reducing the number of π - π stacking structures that can act as exciton quenching sites, it ensures the efficient progression of the Excited-State Proton Transfer (ESPT) process. Second, the suppression of π - π stacking mitigates aggregation-caused quenching (ACQ), thereby preventing the dissipation of ESPT-generated fluorescence via non-radiative pathways"*.

8. The disordered amorphous structure of UPy-DHDI is responsible for the high luminescence, while the crystalline structure of the control sample UPy-DHI leads to weak emission. The authors attribute this to the amorphous structure suppressing exciton annihilation. However, the paper does not provide a detailed mechanistic explanation of why the amorphous state is more favorable for the ESPT process itself. A insightful evidence of how the dynamics of the hydrogen bond network or the conformational freedom in the amorphous state specifically facilitating proton transfer would be offered.

Reply: We thank the reviewer for their valuable comments and insightful suggestion. We have provided further explanation regarding the reason why the amorphous state facilitates the ESPT process. The amorphous state offers a degree of conformational freedom, allowing dynamic

reorganization of the hydrogen-bonding network, which is crucial for the efficient occurrence of ESPT. In Supplementary Figure 14 of the original manuscript, we included solid-state NMR results related to molecular mobility that effectively support this conclusion. The table below lists the ^{13}C T_1 relaxation time measurements, where α , β , and γ represent carbon atoms involved in forming multiple hydrogen-bonding structures. Among the two samples, UPy-DHDI exhibits much shorter ^{13}C T_1 relaxation times, indicating faster molecular motion and higher conformational freedom, which facilitates the proton transfer process and enables highly efficient luminescence. In contrast, UPy-DHI shows significantly longer ^{13}C T_1 times, suggesting restricted molecular mobility that hinders the subtle conformational adjustments required for ESPT. We have discussed these findings in the revised manuscript as highlighted: *“Furthermore, these results provide an explanation for the highly efficient fluorescence of UPy-DHDI from the perspective of conformational freedom. The relatively amorphous structure offers greater conformational flexibility, thereby allowing dynamic rearrangement of the hydrogen-bonding network, which is crucial for the ESPT process”*.

Table R1. T_1 relaxation time of ^{13}C nucleus in UPy core

items	Sample	α	β	γ
$T_1\text{C}$ (s)	UPy-DHI	100	106	129
	UPy-DHDI	46.7	54.7	36.8

9. The paper posits the existence of a second enol form (Enol₂) to explain the long-wave emission peak observed in CHCl_3 solution. While this is supported by theoretical calculations that show a good match between the predicted and experimental wavelengths, this evidence is primarily computational. A significant flaw is the absence of any direct experimental verification of the Enol₂ form's existence, which would substantially strengthen the proposed ESPT mechanism. Further experimental techniques, such as time-resolved spectroscopy specifically designed to probe for this second tautomer, could be necessary.

Reply: We sincerely appreciate the reviewer's comment and insightful suggestion. The direct experimental verification of the Enol₂ form's existence, can indeed enhance credibility of the ESPT mechanism we proposed here. Following the reviewer's suggestion, time-resolved emission

spectroscopy (TRES) was further employed to detect the emission peak from Enol2 configuration, as shown in Figure R10. In the time-resolved emission spectra excited at 405 nm, three distinct emission peaks (1-3) are observed at different decay time, which should be attributed to the keto form and two enol forms. Among them, Peak 3 appears gradually during the decay process, and its relative intensity increases over time, strongly suggesting the existence of the Enol2 configuration. As the delay time increases, the intensities of peaks 1 and 2 gradually weaken, while that of peak 3 gradually strengthens. We hypothesize that the Enol2 forms at a later stage and exists in relatively low abundance, making it challenging to capture its signal in both transient absorption and steady-state fluorescence spectra. Herein, TRES based on time-correlated single photon counting (TCSPC), can capture the weak emission signal from Enol2 with a long delay time once peak 1 and peak 2 have weakened. This supplementary experimental data is crucial for supporting the mechanism we have proposed. Therefore, the relevant discussions have been incorporated into the revised manuscript, as highlighted: *“Time-resolved fluorescence spectroscopy based on time-correlated single photon counting (Supplementary Fig. 12), was employed to capture the weak emission signal from Enol2 with a long delay time. The evolution of the emission spectra with delay time demonstrates the existence of the ESPT process and the existence of the Enol2 form”*.

Figure R10. Time-resolved emission spectrum of PU-UPy.

Reviewer #2 (Remarks to the Author):

The manuscript by Zuo et al deals with unexpected fluorescence properties induced by a well know moiety (ureidopyrimidinone, UPy) within a polymer. A combination of experimental (synthesis, spectroscopies, various applications) and theoretical (TD-DFT, MD) approaches has been convincingly provided to present, rationalize and prove the efficiency and the potential of such systems. Very interestingly, they propose a mechanism involving tautomerization and excited state proton transfer to explain the boost in fluorescence efficiency while the restriction of motions is

often argued to explain it. Please find below some questions and comments I had after reading the manuscript. Some of them should be taken into account to clarify or precise some points within the main text.

Reply: We sincerely appreciate the reviewer's positive comments and the constructive suggestions. A response to all concerns has been systematically address the concerns addressed point-by-point as follows:

1. As no details about the TD-DFT calculations are provided (oscillator strength, nature and description in terms of orbitals/involved states), does the maximum absorption wavelength corresponds to S₁ state, as only the S₁ was optimized to retrieve the emission wavelength? Are HOMO and LUMO the only orbitals involved within the transition?

Reply: We thank the reviewer for pointing this out. The maximum absorption wavelength indeed corresponds to the S₁ state, and the HOMO and LUMO are the sole molecular orbitals involved in this transition.

2.How was the curve of figure 9b in SI obtained? Was it a simple scan (relaxed or not?) with potential energies reported or is it a “real” energy barrier with frequency calculations involved? This should be clarified within the manuscript and SI.

Reply: The curve in Figure SI-9b was obtained through a simple scan (relaxed) of potential energies, and this has been explicitly stated in both the main text and the supplementary materials of the revised manuscript.

3.Does the tautomerization process take place at the ground state (GS), towards the enol form, or only in the excited state (ES)? Is there a competition between the tautomerization at the GS and the ES? In the excited state, following figure 9b in SI, the keto form seems to be more stable than the enol form, is it compatible with the following steps?

Reply: We thank the reviewer for pointing this out. The tautomerization process occurs exclusively in the excited state (ES), as the energy barrier between the keto and enol configurations in the ground state is too high to overcome. We simulated the infrared spectra of three configurations of UPy-DHDI, and the simulated spectrum of the keto form showed excellent agreement with the experimental results, confirming that the ground-state configuration is the stable keto structure (Fig. R11).

Figure R11. (a) FTIR spectra of UPy-DHDI. (b) Simulated FTIR spectra of three configurations of UPy-DHDI: keto, enol1, enol2.

Upon careful re-examination of the results in SI 9b, we identified an error in the original labeling of the x-axis: it should represent the N–H distance rather than the O–H distance. This labeling error has been corrected in the revised manuscript. With this correction, the computational results are consistent with the subsequent processes. We sincerely appreciate the reviewer’s thoroughness and rigor, which helped us rectify this detail and prevent potential misunderstandings.

Figure S9. (a) Optimized structure of UPy-DHDI. (b) The potential energy curve of ESPT product.

4. Are the 4 H-bonds always effective during MD simulations? What are the extra H-bonds observed? Do they correspond to “peripheric” H-bonds? It should be stated that the keto form was considered for MD simulation. Was the MD simulation of the keto form done with the GS or ES structure (if the structures/charges/density repartition are different between GS and ES)? Why the enol1 and enol2 were not considered during MD simulations?

Reply: We sincerely thank the reviewer for these insightful questions. As follows, we will respond to the reviewers point by point:

(1) The quadruple hydrogen bonds (H-bonds) between UPy dimers remained stable and effective throughout the MD simulations. This is supported by the consistently maintained number of H-bonds within the dimer and tetramer systems. Specifically, the dimeric system exhibited between 4 and 6 H-bonds, while the tetrameric system maintained between 8 and 10 H-bonds during the simulation period.

(2) The additional H-bonds beyond the core quadruple set primarily involve intramolecular interactions, specifically between the hydrogen atom in the amino group and the oxygen atom in the carbonyl group of the UPy units (Fig. R12). These can indeed be classified as "peripheric" H-bonds, which contribute additional stability to the supramolecular assembly without disrupting the core quadruple H-bonding network.

(3) The MD simulations were conducted using the ground-state (GS) keto form of the UPy dimer. This form represents the thermodynamically stable configuration under ground state conditions, as further corroborated by the results in Figure R11.

(4) The enol forms (Enol1 and Enol2) are photogenerated species that predominantly exist in the excited state (ES) following excited-state proton transfer (ESPT). Their lifetimes are on the order of picoseconds, as evidenced by our transient absorption spectroscopy data. Conventional MD simulations, which operate on nanosecond to microsecond timescales under thermal equilibrium conditions, are not suitable for capturing these short-lived excited-state tautomer.

We have revised the manuscript and explicitly stating the use of the keto form in the MD simulations, as follows: "MD simulations based on keto form of UPy-DHDI were used to study the aggregated structure of dimers and tetramers, revealing the influence of aggregated structure of chromophores for fluorescent properties."

Figure R12. One structure of UPy-DHDI dimers extracted from the MD simulation, with the red dotted line representing the H-bonds.

5. Why are there such harsh thresholds on the emission spectra of UPy-DHI (figure 7b in SI)? Were the spectra recorded over the entire wavelength range?

Reply: Since the tail of the excitation light may interfere with the fluorescence signal of the sample, a strict threshold cutoff was applied to the detection wavelength to minimize this effect and enable normal detection of the sample's emission signal. However, due to the fact that the emission signal from the UPy-DHI sample is weaker than the tail signal of the excitation light, the aforementioned "harsh thresholds" were employed. The spectrum covers the wavelength range where sample emission is present.

6. Why was hexane used for MD simulations and not DMF or chloroform?

Reply: In molecular dynamics (MD) simulations, the use of a poor solvent is necessary to simulate aggregated states such as dimers or tetramers. In experiments, n-hexane acts as a poor solvent for the system, while DMF and chloroform are good solvents. Therefore, n-hexane was selected as the solvent for the simulations.

7. On figure 2a, the second ESPT appears as reversible (double arrow) but not the first one (simple arrow). Why? Is it due to energetic considerations? This can be added to the figure and/or the text if the authors have the energy barriers or an approximation of it.

Reply: Since our previous ultrafast spectroscopic experiments did not detect signals corresponding to the enol1 \rightarrow enol2 transition, we initially hypothesized that this process might involve ultrafast relaxation and thus represented it with a reversible arrow in the schematic diagram. In the revised manuscript, following reviewer1's suggestion, we successfully detected the enol1-enol2 transition

signal using transient fluorescence spectroscopy (Fig. R10), confirming that the process is irreversible. Accordingly, we have replaced the reversible arrow with an irreversible one in the revised schematic.

8. Are the Enol2 dimers also stable as there are only 2 intermolecular H-Bonds, compared to 4 H-Bonds for the keto/enol1 dimers?

Reply: The enol2 configuration is indeed less stable than enol1 in the excited state, representing an unstable structure, which is consistent with our experimental observations. Although it possesses fewer hydrogen bonds compared to the keto and enol1 configurations, its higher degree of conjugation provides the driving force for the enol1 → enol2 transition.

Typos:

Figure 5: (e) Plots of PL intensity versus temperature (?) for PU-UPy under 365 nm UV light irradiation.

Spectral characterization revealed that UPy-DHI exhibited a quantum yield below 10% and displayed excitation wavelength-dependent fluorescence emission (Supplementary Figs. 10b, d). →

There is no Fig 10b, d in SI.

Figure 12 in SI. Graph b does not represent the emission spectra at different wavelength but at different concentration.

Figure 9b in SI. Unit of the y axis should be kcal mol⁻¹

Reply: We thank the reviewer for the thorough review of our manuscript. The diligent work has brought to our attention several minor errors and oversights in the original version, which greatly contributed to improving the quality of the paper. In the revised manuscript, we have carefully reviewed and corrected the issues mentioned above.